The value of inflammatory indices in the diagnosis of acute appendicitis and prediction of complicated appendicitis: a retrospective study

Yarkaç Akif akifyarkac@hotmail.com 1
Öncü Güldür Çiğdem 1
Bozkurt Seyran 1
Köse Ataman 1
Buyurgan Çağrı Safa 1
Erdoğan Semra 2
Kara Tuba 3
1 Department of Emergency Medicine, Mersin University , Mersin , Turkey
2 Department of Bioistatistics and Medical Informatics, Mersin University , Mersin , Turkey
3 Department of Pathology, Mersin University , Mersin , Turkey
Yilmaz Sarper
Electronic publication date: 2025 Jul 25
Publication date: 2025
Volume: 13
Electronic Location ID: e19754
Received 2025 Feb 14; Accepted 2025 Jun 25
Copyright: ©2025 Yarkaç et al.
Copyright year: 2025
Copyright holder: Yarkaç et al.
License: This is an open access article distributed under the terms of the Creative Commons Attribution License, which permits unrestricted use, distribution, reproduction and adaptation in any medium and for any purpose provided that it is properly attributed. For attribution, the original author(s), title, publication source (PeerJ) and either DOI or URL of the article must be cited.
License URL: https://creativecommons.org/licenses/by/4.0/

Keywords: Acute appendicitis, Complicated appendicitis, Systemic immune-inflammatory index, Systemic inflammation response index, Pan-immune-inflammatory value

Funding: The authors received no funding for this work.

==============================
Background

Acute appendicitis is a common cause of acute abdominal pain, typically treated surgically, although medical management is also an option. Laboratory investigations play a valuable role in diagnosis and predicting prognosis. We aimed to investigate and compare the effectiveness of indices such as the Systemic Immune-Inflammation Index (SII), Systemic Inflammatory Response Index (SIRI), and Pan-Immune-Inflammatory Value (PIV), all derived from complete blood count, in diagnosing acute appendicitis and distinguishing between complicated and uncomplicated cases.

Methods

We retrospectively analyzed data from 334 patients diagnosed with acute appendicitis and 350 healthy individuals. The SII, SIRI, and PIV parameters were compared between the acute appendicitis group and the healthy group. The SII, SIRI, and PIV parameters were also compared between the complicated and uncomplicated groups.

Results

All three parameters demonstrated good performance in predicting the diagnosis of appendicitis. An SII value above 1,008.43 (p < 0.0001, AUC = 0.707), a SIRI value above 2.60 (p < 0.0001, AUC = 0.743), and a PIV value above 904.85 (p < 0.0001, AUC = 0.726) were found to support the diagnosis of appendicitis. For predicting complicated appendicitis, thresholds were identified as SII values above 2501.13 (p = 0.0012, AUC = 0.634), SIRI values above 7.98 (p = 0.0005, AUC = 0.645), and PIV values above 1,869.66 (p = 0.0012, AUC = 0.635).

Conclusions

All three indices showed good predictive performance for diagnosing appendicitis and acceptable performance for predicting complicated cases. Among the indices, SIRI was slightly superior to the other two in both scenarios.

Introduction

Acute appendicitis is a common medical condition, with a lifetime incidence of approximately 7% and a complication rate of up to 20%. It is primarily treated surgically, and diagnosis is based on clinical history and physical examination. Laboratory investigations play an important role in aiding clinicians with diagnosis and prognosis prediction (Cakcak, Türkyılmaz & Demirel, 2022).

Complete blood count (CBC) is a widely accessible, cost-effective laboratory test that provides rapid information on various cell types and morphologic parameters, such as white blood cell count, lymphocyte count, platelet count, and monocyte count. Several indices derived from the CBC are useful in diagnosing and risk stratifying numerous diseases (Siki et al., 2023).

The Systemic Immune-Inflammation Index (SII) is based on neutrophil, lymphocyte, and platelet counts from the CBC, providing insights into the patient’s immune status and inflammation level (Hamad et al., 2022). Similarly, the Systemic Inflammatory Response Index (SIRI), which uses neutrophil, monocyte, and lymphocyte counts, serves as a marker reflecting the balance between immune response and inflammation, offering prognostic information in inflammatory conditions (Chen et al., 2020). The Pan-Immune-Inflammatory Value (PIV), calculated from monocyte, neutrophil, lymphocyte, and platelet counts, is a relatively new marker that has shown promise in providing prognostic data in conditions such as certain cancers and acute myocardial infarction, indicating the patient’s inflammatory status (Murat et al., 2023).

Neutrophils, lymphocytes, monocytes and platelets are blood count components known to be affected by inflammation. The expression of these components as ratios or indices increases the predictive power of the inflammatory state. It has been discussed in the literature that the use of these indices and ratios in events such as autoimmune diseases, cancer types, cardiovascular diseases and infections may provide clearer information about the immune-inflammatory status in the body (Yang et al., 2025; Urbanowicz et al., 2022).

Delayed diagnosis of acute appendicitis can lead to complications, increasing morbidity and mortality. These complications include abscess formation, gangrenous appendicitis, perforation, and phlegmon. Early identification of these complications and timely intervention are critical for reducing adverse outcomes in acute appendicitis (Saridas et al., 2024). The aim of our study is to investigate and compare whether indices such as SII, SIRI, and PIV, derived from complete blood count parameters, can assist clinicians in diagnosing acute appendicitis and differentiating between complicated and uncomplicated appendicitis.

Materials & Methods

No sample size calculation was made for this study. The data of a total of 349 patients who were admitted to our study between 1 January 2021 and 31 December 2023 with abdominal pain and diagnosed as acute appendicitis were analyzed retrospectively. The diagnosis of appendicitis was made as a result of the joint decision of the emergency physician and general surgeon by evaluating clinical signs and symptoms, laboratory parameters, and radiological imaging results together. Among these patients, 15 patients in whom appendicitis was not detected on pathological examination were excluded from the study. Patients with chronic inflammatory disease who were considered to have acute appendicitis were not included in the study. A total of 334 patients were included in the study as the ‘acute appendicitis group.’ Between the same dates, 350 patients who did not have any infective signs and symptoms, inflammatory events, or comorbid diseases were evaluated as the “healthy group”. Patients who had received blood transfusions within the last 1 year (assuming that blood parameters may change), patients with rheumatological diseases, patients with active malignancy and chemotherapy within the last 1 month, and patients with hematological disorders (such as leukemia and lymphoma) were not included in the study.

Post-hoc power analysis was calculated using the G-Power package program. According to the 0.05 Type I error, SII, SIRI, and PIV parameters for different effect sizes (0.565, 0.699, and 0.605) and for 350 healthy and 334 appendicitis groups, the lowest power of the test was calculated as 0.99.

Data collected for the study included age, gender, comorbidities, medications, diagnostic imaging methods, and complete blood count parameters (neutrophil, lymphocyte, monocyte, and platelet counts). For the acute appendicitis group, additional information was recorded regarding whether surgery was performed (open or laparoscopic), the pathology results (acute appendicitis, phlegmonous appendicitis, plastron appendicitis, perforated appendicitis, reactive lymph node, or not appendicitis), the presence of complications according to pathology results, and hospital outcomes (discharge or death). In the healthy group, age, gender, and CBC parameters were similarly recorded on a pre-prepared data form.

Patient data were accessed retrospectively through the hospital’s information management system.

The presence or absence of complications was determined according to the clinical findings, imaging results, and pathology results. Accordingly, patients with phlegmonous or perforated appendicitis were evaluated in the complicated group.

For the diagnosis of appendicitis, the data of the appendicitis group (n = 334) and the healthy group (n = 350) were compared. For the prediction of complications, the data of patients with complicated appendicitis (n = 60) and patients with uncomplicated appendicitis (n = 274) were compared.

The indices were calculated as follows:

• Pan-immune-inflammatory value (PIV) = (neutrophil count × monocyte count × platelet count)/lymphocyte count (Murat et al., 2023)

• Systemic immune-inflammation index (SII) = (platelet count × neutrophil count)/lymphocyte count (Hu et al., 2014)

• Systemic inflammation response index (SIRI) = (neutrophil count × monocyte count)/lymphocyte count (Zhang, Liu & Wang, 2020)

Ethical approval for the study was obtained from the Mersin University Clinical Research Ethics Committee on October 16, 2024, under the approval number 989.

Informed consent was not needed due to the retrospective nature of the study.

Statistical analysis

Normality of continuous variables was assessed using the Shapiro–Wilk test. Comparisons of continuous variables were performed with the Mann–Whitney U test, and results were presented as minimum, maximum, median, and interquartile range. Categorical variables were compared using the Pearson chi-square and likelihood ratio chi-square tests, with results shown as numbers and percentages. Receiver operating characteristic (ROC) analysis was used to determine cut-off points for continuous parameters, with sensitivity, specificity, positive and negative predictive values, and positive and negative likelihood ratio values reported as descriptive statistics. The De Long test was used to test the differences between the ROC curves. Data were analyzed using demo versions of the SPSS v.11.5 (SPSS Inc., Chicago, IL, USA) and MedCalc v.10.3 statistical package programs. The result of bootstrap resampling methods (resampling = 1,000) were obtained from the Jamovi package program. A p-value of < 0.05 was considered statistically significant.

Results

A total of 334 patients in the acute appendicitis group and 350 patients in the healthy group were analyzed. In the appendicitis group, 83.7% of patients underwent open surgery. When evaluated together with the pathology result, it was determined that 60 patients developed complications. The mean age was calculated as 50.4 ± 21.4 in the healthy group, and 40.5 ± 16.6 in the appendicitis group and a statistically significant difference was found between the groups in terms of mean age (p < 0.001). The demographic and medical characteristics of the acute appendicitis group are presented in Table 1.

Table 1 Demographic and medical data of the patients in the acute appendicitis group.

		N	%	
Gender	Male	199	59.6	
Female	135	40.4	
Comorbidity	Heart failure	1	0.3	
Hypertension	2	0.6	
CVA	3	0.9	
DM	11	3.3	
Kidney disease	1	0.3	
Other	40	12	
None	276	82.6	
Drug use	Antibiotics	1	0.3	
NSAID	4	1.2	
Anticoagulants	17	5	
Antiaggregants	11	3.3	
Antiarrhythmics	1	0.3	
None	300	89.9	
Treatment	Open surgery	282	84.4	
Laparoscopic surgery	52	15.6	
Complication	Complicated appendicitis	60	17.2	
Uncomplicated appendicitis	274	78.5	
Notes.

CVA Cerebrovascular Accident

DM Diabetes Mellitus

NSAID Non Steroidal Anti-inflammatory Drugs

The values of SII, SIRI, and PIV were significantly higher in the appendicitis group compared to the healthy group, indicating that all three parameters were effective in diagnosing acute appendicitis (Table 2). Based on ROC analysis, the cut-off values for supporting the diagnosis of appendicitis were identified as follows: SII > 1,008.43 (AUC (95% CI) = 0.707 [0.671–0.740]; p = 0.0001), SIRI > 2.60 (AUC (95% CI) = 0.743 [0.708–0.775]); p = 0.0001) and PIV > 904.85 (AUC (95% CI) = 0.726 [0.692–0.759]; p = 0.0001). SIRI was the most effective parameter in the diagnosis of acute appendicitis. These findings suggest that all three indices performed well in predicting the diagnosis of appendicitis. When the areas under the curve of the 3 parameters are compared, it can be said that the AUC value of the SIRI parameter is higher than the AUC values of both PIV and SII parameters and is statistically significant (difference between areas (95% CI) = 0.0162 (0.00198–0.0304); De Long’s test p = 0.026; difference between areas (95% CI) = 0.0358 [0.012–0.0597]; De Long’s test p = 0.003 respectively). The number of boostrap resampling were set to 1,000 consistenly. Accordingly, similar results were obtained (difference between areas (95% CI) = 0.0162 [0.029–0.003]; De Long’s test p = 0.013; difference between areas (95% CI) = 0.036 [0.013–0.059]; De Long’s test p = 0.002 respectively).

Table 2 Comparison of laboratory parameters of the healthy group and acute appendicitis group.

	Healthy group (n = 350)	Appendicitis group (n = 334)	P	
	Min–Max	Median
(% 25–75)	Min–Max	Median
(% 25–75)		
WBC (×103/µl)	3.26–30.01	10.5 [8.14–13.45]	3.86–30.08	13.49 [10.13–16.43]	<0.001	
Neutrophils (×103/µl)	1.37–27.19	7.00 [5.14–9.95]	2.09–27.45	10.74 [7.62–13.50]	<0.001	
MPV (fL)	8.10–13.50	9.7 [9.2–10.3]	8.30–13.70	10.45 [9.88–11.10]	<0.001	
Monocyte (×103/µl)	0.19–1.69	0.65 [0.51–0.85]	0.03–7.00	0.81 [0.62–1.06]	<0.001	
Lymphocyte (×103/µl)	0.30–8.56	2.07 [1.41–2.71]	0.32–4.78	1.61 [1.16–2.15]	<0.001	
PLT (×103/µl)	21–522	250 [211.8–290.0]	90–569	238 [202–280]	0.029	
RDW (fL)	3.5–22.6	12.9 [12.3–13.8]	11.10–24.50	12.9 [12.3–13.7]	0.992	
CRP (mg/L)	0.1–321.0	3.25 [1.5–6.73]	0.38–537.80	25.76 [7.73–83.11]	<0.001	
PIV	66.72–8,222.95	557.08 [294.91–1,054.31]	67.16–15,540.91	1,201.81 [672.73–2,104.28]	<0.001	
SII	154.26–9,268.36	880.31 [495.0–1,465.2]	167.91–13,282.83	1,498.67 [949.39–2,434.90]	<0.001	
SIRI	0.35–26.61	2.32 [1.22–4.24]	0.33–47.59	5.15 [3.13–9.23]	<0.001	
Notes.

WBC White Blood Cell

MPV Mean Platelet Value

PLT Platelet Count

RDW Red Cell Distribution Width

CRP C-Reactive Protein

PIV Pan-immun-inflammatory value

SII Systemic Immune-Inflammation Index

SIRI Systemic Immune-Inflammation Index

P values written in bold indicate statistical significance.

When the data of patients with complicated appendicitis and patients with uncomplicated appendicitis were compared, it was determined that complicated appendicitis could be predicted at values above 2,501.13 (AUC (95% CI) = 0.634 [0.580–0.686]; p = 0.0012) for SII, 7.98 (AUC (95% CI) = 0.645 [0.591–0.697]; p = 0.0005) for SIRI, and 1,869.66 (AUC (95% CI) = 0.635 [0.581–0.687]; p = 0.0012) for PIV. Accordingly, it can be said that inflammatory indices can predict complicated appendicitis at the cut-off values determined in acute appendicitis patients, and AUC values are close to each other. When the likelihood ratios are analyzed, it can be said that there is a small change between the pre-test and post-test probabilities for the SII parameter for positive results. For negative results, the true intact cannot be distinguished so well in all three parameters. Furthermore, when compared in terms of AUC values, it was found that there was no statistically significant difference between them (De Long’s test p values = 0.441; 0.985 and 0.676, respectively). The bootstrap resampling results were found to be similar (Table 3, Figs. 1, 2).

Table 3 ROC analysis of the effectiveness of inflammatory indices in the diagnosis of appendicitis and prediction of complications.

		Cut-off	AUC
(95% CI)	P	Sensitivity
(95% CI)	Specificity
(95% CI)	PPV
(95% CI)	NPV
(95% CI)	LR(+)
(95% CI)	LR(-)
(95% CI)	
	PIV	>904.85	0.726
(0.692–0.759)	0.0001	64.47
(59.2–69.5)	71.43
(66.4–76.1)	69.2
(63.9–74.2)	66.8
(61.8–71.6)	2.26
(2.0-2.5)	0.5
(0.4-0.6)	
Diagnosis	SIRI	>2.60	0.743
(0.708–0.775)	0.0001	80.80
(76.3–84.8)	56.57
(51.2–61.8)	65.0
(60.3–69.5)	74.7
(69.0–79.8)	1.86
(1.7–2.1)	0.34
(0.3–0.4)	
	SII	>1,008.43	0.707
(0.671–0.740)	0.0001	72.49
(67.5–77.1)	59.43
(54.1–64.6)	64.1
(59.1–68.8)	68.4
(62.9–73.6)	1.79
(1.6-2.0)	0.46
(0.4-0.6)	
	PIV	>1,869.66	0.635
(0.581–0.687)	0.0012	48.33
(35.2–61.6)	75.55
(70.0–80.5)	30.2
(21.3–40.4)	87.0
(82.0–91.0)	1.98
(1.5–2.6)	0.68
(0.5–0.9)	
Complication	SIRI	>7.98	0.645
(0.591-0.697)	0.0005	51.67
(38.4–64.8)	74.45
(68.9–79.5)	30.7
(21.9–40.7)	87.6
(82.6–91.5)	2.02
(1.6-2.6)	0.65
(0.5-0.9)	
	SII	>2,501.13	0.634
(0.580–0.686)	0.0012	43.33
(30.6–56.8)	81.39
(76.3–85.8)	33.8
(23.4–45.4)	86.8
(82.0–90.7)	2.33
(1.7–3.1)	0.70
(0.5–1.0)	
Notes.

AUC Area Under Curve

PPV Positive Predictive Value

NPV Negative Predictive Value

LR(+) Positive Likelihood Ratio

LR(-) Negative Likelihood Ratio

CI Confidence Interval

P values written in bold indicate statistical significance.

Figure 1 ROC analysis of inflammatory indices in predicting the diagnosis of acute appendicitis.

Figure 2 ROC analysis of inflammatory indices in prediction of complicated appendicitis.

Discussion

Acute appendicitis is one of the most common abdominal emergencies worldwide, and early diagnosis and prognosis prediction are crucial for reducing morbidity and mortality (Şener et al., 2023). The inflammatory indices evaluated in this study -SII, SIRI, and PIV- are easily accessible, cost-effective, and straightforward to calculate. Our findings suggest that while all three parameters were effective for diagnosing acute appendicitis, SIRI demonstrated the highest diagnostic performance.

Although these indices were not as successful in predicting complicated appendicitis as they were in diagnosis, they still showed acceptable predictive capability for complications. Among the three, SIRI had the highest area under the curve (AUC) value for detecting complicated cases, suggesting it may be the most reliable parameter for this purpose. Overall, the performance of the indices was relatively close, indicating that all can be useful in clinical practice for assessing the severity of acute appendicitis.

Diagnosing appendicitis can be challenging, often requiring a combination of physical examination, laboratory tests, and imaging modalities. When clinical suspicion arises, ultrasonography and computed tomography (CT) are commonly used as standard diagnostic tools. However, the indiscriminate use of these imaging modalities can lead to unnecessary invasive procedures. For example, surgical intervention on uncomplicated, mildly inflamed appendiceal tissue may be performed instead of opting for medical treatment.

The use of inflammation scores is valuable in the diagnosis and risk classification of appendicitis, especially considering potential risks associated with imaging. CT exposes patients to ionizing radiation, which carries a long-term cancer risk, and ultrasonography may not be available in every emergency department (Andersson et al., 2017). While indices such as SII and SIRI have been discussed in the literature regarding their utility in diagnosing appendicitis and detecting complications, our study is among the first to compare three inflammatory indices -SII, SIRI, and PIV- simultaneously (Cakcak, Türkyılmaz & Demirel, 2022; Şener et al., 2023).

The involvement of platelets, lymphocytes, and neutrophils in the immune response underlies the use of SII as an indicator of inflammation. In the literature, SII has been reported as a predictor of adverse clinical outcomes across various conditions, including oncologic, cardiovascular, and cerebrovascular diseases. Additionally, SII has shown promise in predicting poor outcomes in infectious conditions, such as COVID-19 and sepsis (Liu et al., 2023). Şener et al. (2023) found that SII was effective in diagnosing acute appendicitis and distinguishing between complicated and uncomplicated cases. Interestingly, in their study, SII’s performance in predicting complicated appendicitis was superior to its diagnostic capability (AUC = 0.826). In another study, SII demonstrated effectiveness in diagnosing appendicitis and predicting complications in pregnant patients (Altuğ et al., 2024). Similarly, SII showed good diagnostic performance (AUC = 0.927) in the pediatric population, although its ability to predict complicated appendicitis was lower than its diagnostic power (AUC = 0.646) (Tekeli et al., 2023). In our study, SII was found to be effective for both diagnosing acute appendicitis and predicting complicated cases. However, its performance was lower compared to SIRI and PIV in both areas.

In addition to neutrophils and lymphocytes, monocytes also contribute to inflammation and tumor development. SIRI, which is easily calculated from these three parameters, has been reported as a prognostic predictor in malignancies (Cui et al., 2023). It has also been utilized to assess inflammation levels and prognosis in conditions such as gout, rheumatoid arthritis, ankylosing spondylitis, and systemic lupus erythematosus (Jiang et al., 2023). A study investigating the value of SIRI in diagnosing acute appendicitis and predicting complications found that SIRI may be beneficial for both purposes. In that study, SIRI’s performance in diagnosing acute appendicitis (AUC = 0.716) was superior to its ability to predict complicated appendicitis (AUC = 0.580) (Siki et al., 2023). In alignment with existing literature, our study demonstrated that SIRI was effective in predicting both acute and complicated appendicitis.

PIV is a relatively new inflammation-based biomarker that incorporates neutrophils, lymphocytes, platelets, and monocytes. It has been reported as a useful prognostic test in various cancer types, often outperforming SII, which does not account for monocyte values (Fucà et al., 2020). In patients with rheumatoid arthritis, PIV has proven effective in both diagnosis and distinguishing between remission and active disease (Tutan & Doğan, 2023). Moreover, PIV has been shown to aid in diagnosing and differentiating complicated appendicitis in patients with acute appendicitis, with a performance characterized by a good AUC of 0.761 (Saridas et al., 2024). In our study, PIV demonstrated better performance than SII but was not as strong as SIRI in predicting both the diagnosis of acute appendicitis and complications. However, it can be concluded that the performances of all three parameters–SII, SIRI, and PIV–are closely comparable.

Timely and accurate diagnosis of acute appendicitis is crucial for effective patient management. While the literature contains numerous studies focusing on the diagnosis of appendicitis, there are relatively few that address the prediction of complications. Currently, there is no standardized guideline for detecting complicated appendicitis, though research suggests that such cases require more urgent intervention. In contrast, uncomplicated appendicitis may often be managed with antibiotics alone or may resolve spontaneously without treatment (Bom et al., 2021). No single physical examination finding, laboratory test, or imaging modality should be solely relied upon for the diagnosis of appendicitis or the prediction of its complications. The indices evaluated in our study, when used in conjunction with physical examination, other laboratory tests, and imaging techniques, may aid clinicians in both diagnosing appendicitis and predicting associated complications. Our study is significant in that it compares three different inflammatory indices for both the diagnosis of acute appendicitis and the prediction of complications, providing meaningful results that contribute to the existing body of knowledge.

Conclusion

Physical examination and laboratory tests are crucial for diagnosing acute appendicitis. Inflammatory indices derived from hemogram parameters, such as SII, SIRI, and PIV, are inexpensive, simple, and easily accessible tools that can assist clinicians in managing patients with appendicitis. All three indices demonstrated good performance in diagnosing appendicitis and were acceptably effective in predicting complicated cases. Among them, SIRI showed slightly superior performance for both conditions.

Limitations

This study has notable limitations, including its retrospective design and single-center nature. Due to the retrospective design of the study, we could not take symptom durations into account and could not make comparisons with the classical scores accepted in the literature. The fact that conditions such as smoking and obesity, which may affect the inflammation status, could not be evaluated may have affected the results of our study.

Additionally, the relatively small sample size may affect the generalizability of our findings. Therefore, multicenter, prospective studies involving larger patient populations are warranted to validate these results.

Supplemental Information

Supplemental Information 1 Raw data

Supplemental Information 2 Healthy group data

Additional Information and Declarations

Competing Interests

Author Contributions

Human Ethics

Data Availability

Clinical Trial Registration

The authors declare there are no competing interests.

Akif Yarkaç conceived and designed the experiments, analyzed the data, prepared figures and/or tables, authored or reviewed drafts of the article, and approved the final draft.

Çiğdem Öncü Güldür conceived and designed the experiments, analyzed the data, prepared figures and/or tables, and approved the final draft.

Seyran Bozkurt performed the experiments, analyzed the data, prepared figures and/or tables, and approved the final draft.

Ataman Köse conceived and designed the experiments, performed the experiments, prepared figures and/or tables, and approved the final draft.

Çağrı Safa Buyurgan conceived and designed the experiments, performed the experiments, analyzed the data, authored or reviewed drafts of the article, and approved the final draft.

Semra Erdoğan analyzed the data, authored or reviewed drafts of the article, and approved the final draft.

Tuba Kara conceived and designed the experiments, performed the experiments, analyzed the data, authored or reviewed drafts of the article, and approved the final draft.

The following information was supplied relating to ethical approvals (i.e., approving body and any reference numbers):

This research was approved by Mersin University Clinical Research Ethics Committee on October 16, 2024. (989).

The following information was supplied regarding data availability:

The raw data is available in the Supplemental Files.

The following information was supplied regarding Clinical Trial registration:

213212313.

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
