# Peer review of "The value of inflammatory indices in the diagnosis of acute appendicitis and prediction of complicated appendicitis: a retrospective study"

_PeerJ, doi:10.7717/peerj.19754_

## Round 0.1 · original submission · Major Revisions

Dear Authors,

I think it will contribute to your article scientifically in line with your referee suggestions about your article.

·

Basic reporting

Clear, concise and understandable English is used in the article.
The references used in the article appear to be up-to-date and relevant to the topic and content of the article.
The tables and images used in the article are sufficiently descriptive and adequately summarize the article's conclusions.
The aim and hypothesis of the article are appropriate and the results obtained do not contradict the hypothesis and purpose.
When the article is examined in general, it is seen that it has sufficient academic content and level.
As the authors also mentioned, acute appendicitis is a very common disease that requires surgical treatment. Its diagnosis is based on physical examination and the patient's history.
With the increase in imaging methods in recent years, additional methods are often needed to confirm the diagnosis in order to reduce the negative surgical rate.
Although imaging methods are increasing and widely used, it is not easy to access imaging methods all over the world.Even in developed countries, it may not always be possible to access imaging methods in all parts of the country.
In addition, imaging methods such as CT are not innocent because they contain radiation. In this process, the use of laboratory methods to support diagnosis is widely investigated. The use of additional methods that do not have side effects such as radiation and require less economic costs will strengthen the clinician's hand for diagnosis.

Experimental design

The design of the study, which is retrospective in nature, seems appropriate to question the hypothesis and obtain valid data.
The study complied with research ethics rules and received ethical approval from the local institution. In addition, there is no ethical problem due to the nature of the research.
The study method, patient selection and statistical analysis are stated adequately and descriptively.

Validity of the findings

No bias findings were detected in the research data and statistical data are appropriate and valid.
The data of the research are presented in an appropriate order and in an understandable and clear manner.

Additional comments

As I mentioned above, although it is a retrospective descriptive study, my opinion is positive due to the compliance with the methodology, the validity of the findings and their relevance in clinical use.

Reviewer 2 ·

Basic reporting

Clear understandable and unambiguous, English used in the article.
The references used in the article are sufficient, up-to-date and relevant to the article content.
The tables and figures in the article are sufficient and provide a summary of the results of the study.
The aim and hypothesis of the article appropriate . The results of the article do not contradict the hypothesis and purpose.

Experimental design

Retrospective design, valid data in accordance with the hypothesis were obtained.
The study method, patient selection and statistical method used were well defined and appropriate.
The article presents data on the use of easily accessible laboratory tests in the emergency department without or in support of radiological imaging in the diagnosis of appendicitis and complicated appendicitis.
It was observed that the article was approved by the ethics committee and prepared in accordance with ethical rules.

Validity of the findings

A study with appropriate statistical data, well-expressed results, and that could benefit the literature.

Additional comments

I think the article contains useful information for emergency medicine readers.

Reviewer 3 ·

Basic reporting

1) The manuscript is written in clear and professional English, with minor areas that could benefit from improvement in clarity.
2) The introduction adequately presents the rationale for the study, stating that inflammatory indices may aid in diagnosing acute appendicitis and predicting complications. However, a more detailed discussion on how these indices have been used in previous studies and what other approaches have been employed for similar classifications would enhance the background.
3) Figures and tables are relevant and appropriately labeled. However, the raw dataset appears incomplete, as the dataset containing healthy controls was not accessible for verification.
4) The study follows a professional structure and adheres to PeerJ's standards.
5) Literature references are appropriate and relevant to the study context.

Experimental design

1)The research falls within the journal's scope and presents an original investigation.
2)The methodology clearly defines two patient groups: one with acute appendicitis and another consisting of asymptomatic healthy individuals. However, the study design is not explicitly stated, and the selection criteria for the healthy group should be clarified. Specifically, it is unclear from which population these individuals were recruited and whether they underwent any screening procedures.
3)The exclusion of patients receiving blood transfusions is noted, but the rationale for this exclusion should be explicitly stated.
4)Complications were determined based on pathology rather than clinical findings. The justification for this approach should be further elaborated, particularly how it aligns with clinical practice.
5)The ethical approval section states that an informed consent waiver was granted due to the retrospective nature of the study. This information should be explicitly mentioned in the methodology section for transparency.
6) The inflammatory indices (SII, SIRI, PIV) are all derived from similar blood parameters. Has multicollinearity been assessed among these predictors? If two or more indices provide redundant information, then their independent contribution to the diagnostic process is questionable. Variance inflation factor (VIF) analysis or principal component analysis (PCA) could be useful to determine whether these indices genuinely offer distinct predictive value.

7) The study presents diagnostic indices but does not test their reproducibility on an independent dataset. Was an internal validation technique such as bootstrapping performed, or was an independent test set used? Without such validation, the reported diagnostic performance may be overly optimistic due to sample-specific biases.

8) The manuscript interprets diagnostic performance purely from a frequentist perspective (AUC, sensitivity, specificity). However, in real-world clinical decision-making, the pre-test probability of disease heavily influences the utility of a test. Have Bayesian statistics or likelihood ratio analyses been used to contextualize the reported test performance in a realistic clinical scenario?

Validity of the findings

1) The study does not adequately account for potential confounders that could impact inflammatory responses, such as chronic inflammatory diseases, smoking, or obesity. These factors should be controlled or at least discussed as limitations.

2) The retrospective design introduces biases in patient selection and data quality. The study does not specify how patients were recruited, whether their diagnostic processes followed a standardized protocol, or how the control group was selected. This raises concerns about data reliability and potential selection bias.

3) The clinical course of patients with acute appendicitis is not considered. The study does not evaluate symptom duration or clinical scores like the Alvarado or AIR score, which are crucial for assessing disease severity and guiding management. Additionally, the diagnostic performance of inflammatory indices is not compared to existing clinical scores, making it unclear whether they provide additional value beyond traditional assessment tools.

4) The study does not account for how inflammatory markers change over time. The timing of sample collection in relation to symptom onset could significantly affect biomarker levels, but this is not analyzed.

5) The differences in AUC values between indices are small (e.g., 0.707 vs 0.743), yet no statistical comparison is performed to determine whether these differences are clinically meaningful. A DeLong test or an equivalent method should be conducted to determine statistical significance between AUC values.

6) The rationale for the selection of cutoff values in ROC analysis is unclear. It is important to state whether these values were derived using Youden's index or another method.

7) The statistical methods used for normality testing should be reconsidered. The manuscript states that the Shapiro-Wilk test was used; however, given the sample size exceeding 50, visual assessment via histograms is also recommended. The presented data suggest that normality assumptions were not met, indicating that alternative statistical approaches may be warranted. Furthermore, if means and standard deviations are used despite non-normality, mean differences and 95% confidence intervals should be reported, and this approach should be justified in the discussion section.

8) The discussion section asserts that inflammatory indices may help in assessing the severity of acute appendicitis. However, since the comparison was made against a completely healthy control group rather than between uncomplicated and complicated appendicitis cases, the real-world applicability of these findings remains uncertain. The manuscript also does not compare these indices to widely used risk scores such as Alvarado, missing an opportunity to discuss their clinical utility.

9) Conducting multiple statistical tests increases the probability of Type I errors (false positives). The manuscript should address whether adjustments for multiple comparisons were made to control the family-wise error rate.

10) The study does not provide a clear rationale for the chosen sample size. An a priori power analysis should be conducted to ensure the study is adequately powered to detect clinically meaningful differences.

Additional comments

1)The first paragraph of the discussion states that SIRI had the highest diagnostic performance. However, the results section does not present a formal AUC comparison.
2)The manuscript implies that these inflammatory indices can be used to predict appendicitis severity, yet no specific recommendation is given on their use in clinical practice.
3)In lines 138-140, the manuscript claims that these indices may be useful in clinical decision-making, but given that all comparisons were made with healthy controls, the real-world application of this claim is unclear.
4)The discussion should be expanded to consider the potential limitations of relying solely on inflammatory indices in diagnosing and managing appendicitis.
5)Overall, while the study presents an interesting approach, further refinements in statistical analysis and discussion of clinical applicability would improve the manuscript. Additionally, the study lacks a dedicated limitations section, which should be included to acknowledge constraints such as its retrospective design and potential selection biases.

Reviewer 4 ·

Basic reporting

Revision Suggestions for Authors
Dear Authors, your work provides valuable information on inflammatory indices for appendicitis diagnosis and prognosis. To meet PeerJ standards, please address the following:

1. Data Transparency: Define units for laboratory parameters in tables (e.g. neutrophils ×10⁹/L) and ensure that ROC curve axes are fully labeled.

2. Language Polishing: Attempt to correct minor grammatical inconsistencies.

These revisions will improve reproducibility and clinical applicability. We recommend that you resubmit after addressing these points.

Suggestion: Minor Revision

Experimental design

Revision Suggestions for Authors
Dear Authors, your work provides valuable information on inflammatory indices for appendicitis diagnosis and prognosis. To meet PeerJ standards, please address the following:

1. Data Transparency: Define units for laboratory parameters in tables (e.g. neutrophils ×10⁹/L) and ensure that ROC curve axes are fully labeled.

2. Language Polishing: Attempt to correct minor grammatical inconsistencies.

These revisions will improve reproducibility and clinical applicability. We recommend that you resubmit after addressing these points.

Suggestion: Minor Revision

Validity of the findings

Revision Suggestions for Authors
Dear Authors, your work provides valuable information on inflammatory indices for appendicitis diagnosis and prognosis. To meet PeerJ standards, please address the following:

1. Data Transparency: Define units for laboratory parameters in tables (e.g. neutrophils ×10⁹/L) and ensure that ROC curve axes are fully labeled.

2. Language Polishing: Attempt to correct minor grammatical inconsistencies.

These revisions will improve reproducibility and clinical applicability. We recommend that you resubmit after addressing these points.

Suggestion: Minor Revision

Additional comments

Revision Suggestions for Authors
Dear Authors, your work provides valuable information on inflammatory indices for appendicitis diagnosis and prognosis. To meet PeerJ standards, please address the following:

1. Data Transparency: Define units for laboratory parameters in tables (e.g. neutrophils ×10⁹/L) and ensure that ROC curve axes are fully labeled.

2. Language Polishing: Attempt to correct minor grammatical inconsistencies.

These revisions will improve reproducibility and clinical applicability. We recommend that you resubmit after addressing these points.

Suggestion: Minor Revision

---

## Round 0.2 · Major Revisions

Dear Authors,

I am pleased to see that you have successfully addressed many of the suggestions provided by other reviewers. However, I would like to highlight that one of the reviewers, who clearly recognizes the potential contribution of your study to the literature and acknowledges its overall quality, aims to further enhance your manuscript academically.

I believe their comments will provide valuable insights and contribute meaningfully to the improvement of your work. Therefore, I kindly ask you to read the suggestions carefully and revise your manuscript accordingly.

Reviewer 3 ·

Basic reporting

The manuscript is written in clear, professional English. The introduction has been improved, and the background now better outlines the relevance of inflammatory indices in appendicitis. Literature references are appropriate, current, and relevant. Figures and tables are of good quality, and the raw dataset has been made available.
The article is self-contained and structured appropriately with standard sections. All appropriate data are now shared in line with the data sharing policy. The manuscript presents a coherent unit of publication and does not appear to have been unnecessarily subdivided.

Experimental design

The study presents an original retrospective analysis of inflammatory indices in acute appendicitis. The research question remains relevant, but critical issues with the study design persist:
The study still compares appendicitis patients to healthy individuals, which is methodologically incompatible with a diagnostic accuracy study aiming for clinical utility.
→ If the goal is diagnostic performance, the authors must compare against patients with alternative causes of abdominal pain. Otherwise, the results only demonstrate physiological differences, not diagnostic discriminative capacity.
→ To illustrate the magnitude of this issue: If the only comparator is a healthy group, even the symptom of "having abdominal pain" would yield an AUROC of 1, as it perfectly separates the groups by design. This renders any additional diagnostic metric (like inflammatory indices) inherently inflated and clinically uninformative.
→ The authors must clearly and honestly redefine the aim as a physiological evaluation or fully acknowledge this critical design limitation and adjust their claims accordingly.
The authors justify not performing internal validation due to the retrospective, exploratory nature of the study. This is unacceptable for a study reporting diagnostic performance.
→ At a minimum, internal validation (e.g., bootstrapping, cross-validation) must be performed and reported to avoid optimism bias and overfitting.
The authors added p-values for DeLong comparisons between AUCs, but they did not report the absolute difference in AUC with 95% CI, which is best practice.
→ This is not strictly a fatal flaw but should be corrected for completeness and clarity. I recommend the authors report the absolute AUC differences with 95% CIs alongside the p-values to allow meaningful interpretation.

Validity of the findings

While the raw data have been made available, the underlying analyses are not yet statistically sound due to the above flaws:
95% CIs are missing for all AUROCs, sensitivities, specificities, and likelihood ratios.
→ This is a reporting deficiency that must be corrected to align with accepted diagnostic study standards (e.g., STARD).

Additional comments

The conclusions, while mostly consistent with the presented data, still overreach by implying diagnostic utility without appropriate study design or validation. Claims of clinical applicability should be substantially toned down, or the aims reframed.

Reviewer 4 ·

Basic reporting

Clear, concise, professional English is used throughout.
Literature references, sufficient field background/context are provided.
Professional article structure, figures, tables contribute to the literature.
It provides results that are consistent with the hypotheses.
It is appropriate to publish the article in this form.

Experimental design

Original primary research within the Journal's Aim and Scope.
The research question is well defined, relevant, and meaningful. The research fills an identified knowledge gap in the literature.
Rigorous work performed in accordance with high technical and ethical standards.
Methods are described in sufficient detail and informatively to be replicable.

Validity of the findings

I believe it will contribute to the literature.
All the basic data is provided; it is robust, statistically sound and controlled.
The results are well stated, related to the original research question and limited to supporting conclusions.

---

## Round 0.3 · accepted · Accept

Dear Authors,

I commend your efforts throughout the process. I believe that, together with our reviewers, we have helped advance your manuscript to a stronger position.

Reviewer 3 ·

Basic reporting

The authors have addressed my concerns and suggestions. I have no further critics.

Experimental design

The authors have addressed my concerns and suggestions. I have no further critics.

Validity of the findings

The authors have addressed my concerns and suggestions. I have no further critics.